# BOUND-AND-AVERAGE: LEVERAGING WEIGHTS AS KNOWLEDGE FOR CLASS INCREMENTAL LEARNING

## ABSTRACT

We present a novel training approach, named Bound-and-Average (BaA) for Class Incremental Learning (CIL) that leverages weight ensemble and constrained optimization, motivated by recent advances in transfer learning. Our algorithm contains two types of weight averaging: *inter-task weight averaging* and *intra-task weight averaging*. Inter-task weight averaging integrates the ability of previous models by averaging the weights of models from all previous stages. On the other hand, intra-task weight averaging enriches the learning of current task by averaging the model parameters within current stage. We also propose a bounded update technique that aims to optimize the target model with minimal cumulative updates and preserve knowledge from previous tasks; this strategy reveals that it is possible to effectively obtain new models near old ones, reducing catastrophic forgetting. BaA seamlessly integrates into existing CIL methods without modifying architecture components or revising learning objectives. We extensively evaluate our algorithm on standard CIL benchmarks and demonstrate superior performance compared to state-of-the-art methods.

## 1 INTRODUCTION

Despite the remarkable achievements of recent deep neural networks (DNNs) (Radford et al., 2021; Ho et al., 2020; Brown et al., 2020), training under continually shifting data distributions encounters a significant challenge called *catastrophic forgetting*—performance degradation on previously learned data. Since the real-world environments, in which DNNs are deployed, diversely and dynamically change over time, addressing this issue becomes pivotal for enhancing the efficiency and applicability of DNNs.

Class Incremental Learning (CIL) (Douillard et al., 2020; Hou et al., 2019; Rebuffi et al., 2017; Simon et al., 2021; Kang et al., 2022) is a continual learning framework which learn a sequential influx of tasks composed of disjoint class sets. Given the constraint of permitting only a few or no data from previous tasks during training for new tasks, catastrophic forgetting becomes a significant obstacle for CIL. Prior approaches have attempted to tackle this issue through methods including knowledge distillation (Hinton et al., 2015; Romero et al., 2015; Zagoruyko & Komodakis, 2017; Kang et al., 2022), architecture expansion (Rusu et al., 2016; Yoon et al., 2018; Liu et al., 2021; Yan et al., 2021; Abati et al., 2020), or parameter regularization (Aljundi et al., 2018; Kirkpatrick et al., 2017; Zenke et al., 2017). However, these methods have inherent limitations, such as reliance on data from previous tasks for distilling knowledges of previous tasks, the need for additional network components, and poor performance, which hinder their wide-range applications.

We propose a novel training approach for CIL, referred to as Bound-and-Average (BaA), which can be easily incorporated into existing CIL methods without any algorithmic or architecture modifications. Motivated by recent studies (Wortsman et al., 2022a; Rame et al., 2022; Wortsman et al., 2022b; Izmailov et al., 2018), which demonstrates the effects of weight averaging in aggregating the ability of multiple models, we introduce two types of weight averaging tailored for CIL: inter-task weight averaging and intra-task weight averaging, each respectively enhances the stability and plasticity of a CIL model. To preserve all the knowledge acquired up to the current task and avoid an over-reliance on the data from immediately preceding tasks, we introduce *inter-task weight averaging*. This method summarizes the knowledge acquired throughout the previous tasks by averaging the parameters of the models learned from individual incremental stages in an online fashion, thus

forming a base model. The base model serves as the initialization point for the subsequent task. On the other hand, we propose *intra-task weight averaging* to enhance the model's adaptivity to new tasks. This technique improves the model's generalization capability for each new task by averaging multiple checkpoints along the training trajectory within the current task.

Furthermore, we incorporate *bounded model update* (Tian et al., 2023; Gouk et al., 2020) for training in each task, which constrains weight updates in the vicinity of the base model. By preventing the model parameters from deviating excessively from the base model, this strategy ensures stability and preserves knowledge from previous tasks. Throughout integrating these techniques, we strike a balance between stability, preserving knowledges from previous tasks, and plasticity, adapting to new tasks in CIL scenarios.

The contributions of this paper are summarized as follows:

- We incorporate two weight averaging techniques into CIL, inter- and intra-task weight averaging. These techniques enhance the model's stability and plasticity by averaging model parameters across tasks (inter-task) and within a task (intra-task), respectively.

- We introduce a bounded model update strategy that constrains the total amount of model updates within each task. By enforcing the new models to remain close to the old ones, the proposed technique alleviates the catastrophic forgetting of previously acquired knowledge.

- Our algorithm, which can be conveniently integrated into existing CIL methods, consistently improves performance on multiple benchmarks with marginal extra computational complexity. We demonstrate the effectiveness of our method via extensive experiments.

## 2 RELATED WORKS

This section reviews existing algorithms related to class incremental learning and transfer learning.

### 2.1 CLASS INCREMENTAL LEARNING (CIL)

CIL is a challenging problem that aims to learn a model with the number of classes increasing stage-by-stage without forgetting the previously learned classes. We organize CIL methods into five groups based on their main strategies: parameter regularization, architecture expansion, bias correction, knowledge distillation, and rehearsal methods.

Parameter regularization methods (Aljundi et al., 2018; Kirkpatrick et al., 2017; Zenke et al., 2017) measure the importance of network parameters and adjust their flexibility to mitigate catastrophic forgetting. However, these methods suffer from unsatisfactory generalization performance in CIL scenarios (van de Ven & Tolias, 2019; Hsu et al., 2018). Architecture expansion methods (Rusu et al., 2016; Yoon et al., 2018; Liu et al., 2021; Yan et al., 2021; Abati et al., 2020) dynamically expand the network capacity to handle incoming tasks by adding new neurons or layers. However, they introduce computational burdens due to additional network components. Bias correction methods (Hou et al., 2019; Wu et al., 2019) address the bias towards new classes caused by the class imbalances in CIL by introducing scale and shift parameters or matching the scale of weight vectors. Knowledge distillation methods (Hinton et al., 2015; Romero et al., 2015; Zagoruyko & Komodakis, 2017; Kang et al., 2022) encourage models to preserve previous task knowledge by mimicking the representations of old models. Several approaches match output distributions or attention maps to preserve important information. Rehearsal-based methods (Rebuffi et al., 2017; Ostapenko et al., 2019; Shin et al., 2017) store representative examples or employ generative models to mitigate forgetting. Examples include maintaining class centroids Rebuffi et al. (2017) or using generative adversarial networks (Goodfellow et al., 2014; Liu et al., 2020a; Odena et al., 2017) to generate synthetic examples.

In contrast, we propose a novel approach which is compatible and easy to integrate with the existing CIL methods, without requiring any changes to the network architectures or loss functions. Our approach leverages the weights of previous models instead of the data from previous tasks, which reduces the data dependency on previous tasks. Also, our approach has a negligible computational overhead, making it suitable for various CIL scenarios.

## 2.2 Robust Transfer Learning

Robust transfer learning aims to adapt a pre-trained model to a new domain or task without losing its generalization ability. This is similar to continual learning, which seeks to prevent catastrophic forgetting of previous knowledge while learning new tasks sequentially.

A common approach to robust transfer learning is to freeze or constrain the weights of the pre-trained model (Kumar et al., 2022; Lee et al., 2023) while fine-tuning. This prevents feature distortion from adaptation and preserves the original knowledge. For example, LP-FT (Kumar et al., 2022) adopts a two-step strategy: first, train the linear classifier and then fully fine-tune the entire network with a small learning rate. Surgical fine-tuning (Lee et al., 2023) shows that training only a subset of layers while freezing others can improve the robustness against distribution shift.

Another line of work explores the idea of weight averaging or interpolation to enhance the performance of fine-tuned models (Neyshabur et al., 2020; Wortsman et al., 2022a; Ramé et al., 2022). Neyshabur *et al.* (Neyshabur et al., 2020) shows that interpolating the weights of fine-tuned models from the same pre-trained model can improve both in-distribution accuracy and out-of-distribution robustness. Model Soups (Wortsman et al., 2022a) and Model Ratatouillie (Ramé et al., 2022) extend this idea by using different hyperparameters and downstream tasks to create diverse fine-tuned models and maximize the ensemble effect. WiSE-FT (Wortsman et al., 2022b) proposes to linearly interpolate the weights of the pre-trained model and the fine-tuned model, which allows the model to learn new information without deviating too much from the original network. MARS-PGM (Gouk et al., 2020) and TPGM (Tian et al., 2023) adopt a similar weight projection technique to regularize the fine-tuned models. However, these methods primarily concentrate on a single downstream task, neglecting the sequential domain shifts that are inherent in continual learning.

Stojanovski *et al.* (Stojanovski et al., 2022) introduced Momentum-based Weight Interpolation for Continual Learning (MCL), a method that uses exponential moving average (EMA) of the weights to preserve previous knowledge. However, MCL does not account for the dynamics of continual learning and simply applies the techniques from transfer learning. We address this limitation by investigating the effects and impacts of linear interpolation in continual learning dynamics through inter-task, intra-task weight averaging, and bounded updates on weight space.

## 3 Proposed Approaches

This section presents the details of the proposed method, Bound-and-Average (BaA), for preserving knowledge from previous tasks while learning new tasks in CIL through weight averaging and bounded model update.

### 3.1 Problem Formulation

CIL is a learning framework to handle a sequence of tasks, $\mathcal{T}_{1:K} = \{T_1, \cdots, T_k, \cdots, T_K\}$. Each task $T_k$ consists of a labeled dataset $\mathcal{D}_k$ whose label set, $\mathcal{C}_k$, is disjoint to the ones defined in the past, *i.e..* $(\mathcal{C}_1 \cup \cdots \cup \mathcal{C}_{k-1}) \cap \mathcal{C}_k = \emptyset$. At the $k^{\text{th}}$ incremental stage, the current model $M_k(\cdot)$ is trained on integrated dataset $\mathcal{D}'_k = \mathcal{D}_k \cup \mathcal{B}_{k-1}$, where $\mathcal{B}_{k-1}$ is a memory buffer for storing representative exemplars that belong to all the previously learned classes. The performance of a CIL algorithm is evaluated using a test set comprising test data from all the stages.

### 3.2 Weight Averaging

We propose two unique weight averaging techniques: Inter-task weight averaging and intra-task weight averaging. Although both techniques concentrate on integrating the competency of multiple models, they have distinct objectives. Inter-task weight averaging aims to consolidate knowledge from *all previously learned tasks* and construct a comprehensive model by computing moving averages of previous models. Meanwhile, intra-task weight averaging, motivated by SWA (Izmailov et al., 2018), is effective for boosting the model's generalization capability on *new tasks*. This is achieved by calculating the average weight of models at various checkpoints along the training trajectories. Note that, for both inter- and intra-task weight averaging, it is not required to store all the previous models for averaging because the model averaging is performed in an online manner.

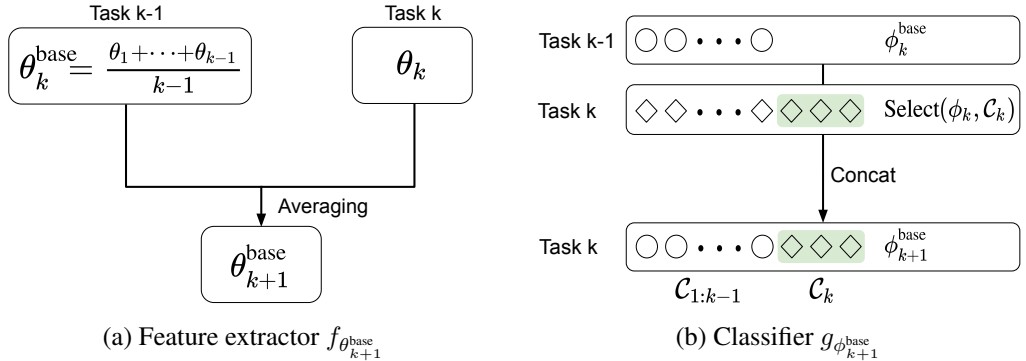

(a) Feature extractor $f_{\theta_{k+1}^{\text{base}}}$

(b) Classifier $g_{\phi_{k+1}^{\text{base}}}$

Figure 1: Description of inter-task weight averaging: Upon the completion of the $k^{\text{th}}$ incremental stage, we establish the base model $M_{k+1}^{\text{base}}(\cdot)$, which will serve as the *initialization point* for the next $(k+1)^{\text{th}}$ stage. The model comprises a feature extractor $f_{\theta_{k+1}^{\text{base}}}(\cdot)$ and a classifier $g_{\phi_{k+1}^{\text{base}}}(\cdot)$, constructed as follows: (a) To construct the base feature extractor $f_{\theta_{k+1}^{\text{base}}}(\cdot)$, we set $\theta_{k+1}^{\text{base}}$ to the moving average of all previous feature extractor weights $\theta_1, \theta_2, \cdots, \theta_k$, which can be easily computed with $\theta_k^{\text{base}}$ and $\theta_k$ following Equation 1. (b) For the classifier $g_{\phi_{k+1}^{\text{base}}}(\cdot)$, we concatenate the weights of the current base classifier $\phi_k^{\text{base}}$ with the weights of the current classifier $\phi_k$ associated with the class set of the current task $\mathcal{C}_k$.

### 3.2.1 INTER-TASK WEIGHT AVERAGING

Let the base model, $M_k^{\text{base}}(\cdot)$, be an average of the models for all the preceding tasks, denoted by $\{M_1(\cdot), \cdots, M_{k-1}(\cdot)\}$. The base model is composed of a feature extractor $f_{\theta_k^{\text{base}}}(\cdot)$ and a classifier $g_{\phi_k^{\text{base}}}(\cdot)$, collectively parameterized by $\Theta_k^{\text{base}} = \{\theta_k^{\text{base}}, \phi_k^{\text{base}}\}$.

Upon the completion of the $k^{\text{th}}$ incremental stage, we combine the current feature extractor $f_{\theta_k}(\cdot)$ and classifier $g_{\phi_k}(\cdot)$ into the current base model, $M_k^{\text{base}}(\cdot)$, as illustrated in Figure 1. This process creates the new base model for the next stage, denoted by $M_{k+1}^{\text{base}}(\cdot)$. The new feature extractor is given by the moving average of all models as follows:

$$\theta_{k+1}^{\text{base}} = \frac{k-1}{k} \cdot \theta_k^{\text{base}} + \frac{1}{k} \cdot \theta_k. \tag{1}$$

To obtain the classifier of the new base model for the $(k+1)^{\text{st}}$ stage, we concatenate the current base model classifier $g_{\phi_k^{\text{base}}}(\cdot)$ defined for the classes in $\mathcal{C}_{k-1}$ with the weights corresponding to the classes in $\mathcal{C}_k$ of the current classifier, $g_{\phi_k}(\cdot)$, which is expressed as

$$\phi_{k+1}^{\text{base}} = \text{Concat}(\phi_k^{\text{base}}, \text{Select}(\phi_k, \mathcal{C}_k)), \tag{2}$$

where $\text{Select}(\phi_k, \mathcal{C}_k)$ extracts the weights corresponding to the classes in $\mathcal{C}_k$ of the current classifier $g_{\phi_k}(\cdot)$. At the $(k+1)^{\text{st}}$ incremental stage, we start to train from the base model, $M_{k+1}^{\text{base}}(\cdot)$, which encapsulates all the knowledge learned up to the current stage.

### 3.2.2 INTRA-TASK WEIGHT AVERAGING

For intra-task weight averaging, we seek to enhance generalization ability on the current task by averaging multiple checkpoints along the training trajectories. In the $k^{\text{th}}$ incremental stage, an intra-task averaged model, $M_k^{\text{avg}}(\cdot)$, parameterized by $\Theta_k^{\text{avg}}$ is updated at every $e_a$ epochs as

$$\Theta_k^{\text{avg}} \leftarrow \frac{n \cdot \Theta_k^{\text{avg}} + \Theta_k}{n+1}, \tag{3}$$

where $n$ denotes the number of models involved in the averaging. Once the stage is completed, the final model, $M_k(\cdot)$, is replaced by the intra-task averaged model, $M_k^{\text{avg}}(\cdot)$, *i.e.*, $\Theta_k \leftarrow \Theta_k^{\text{avg}}$, for inference as well as the computation of the base model, $M_{k+1}^{\text{base}}(\cdot)$.

For the models equipped with Batch Normalization (BN) (Ioffe & Szegedy, 2015), an additional data pass is required for post-training to compute the new running estimates of mean and variance of

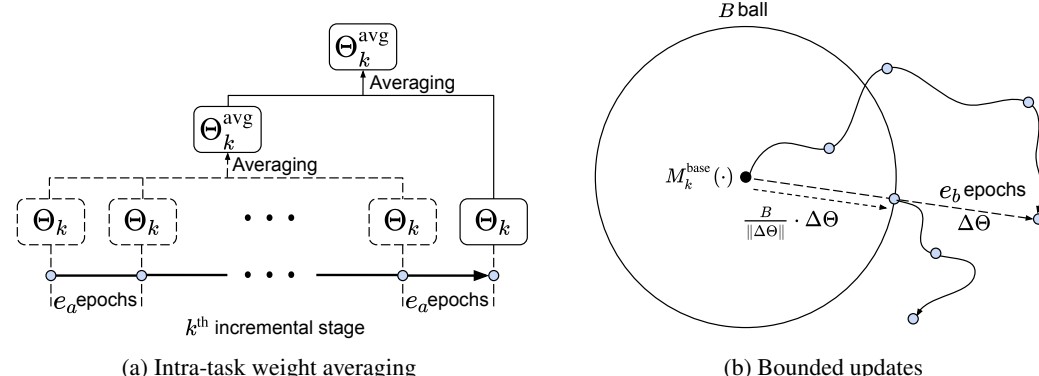

(a) Intra-task weight averaging          (b) Bounded updates

Figure 2: (a) Explanation of the process of intra-task weight averaging: We define an intra-task averaged model weight to compute the running mean of multiple weights along the training trajectories, as described in Equation 3. This averaged model weight is utilized for inference and for computing the next stage base model $M_k^{\text{base}}(\cdot)$. (b) Description of the bounded update method: We constrain the weight updates around the given base model. This strategy is designed to preserve the knowledge embedded in the base model, denoted as $M_k^{\text{base}}(\cdot)$.

the activations after model averaging (Garipov et al., 2018; Izmailov et al., 2018), since the accumulated BN statistics are not computed for the averaged models. Rather than executing an additional forward pass after resetting the running statistics as in SWA (Izmailov et al., 2018), we conduct an additional forwarding path with the running statistics of the current model $M_k(\cdot)$ estimated before model averaging. This strategy alleviates the bias towards the current task due to sample deficiency of the classes introduced in the previous tasks.

### 3.3 BOUNDED MODEL UPDATE

Recent advancements in transfer learning research (Tian et al., 2023; Gouk et al., 2020) underscores the significance of confining the divergence of fine-tuned models from the pretrained models. This strategy allows the fine-tuned model to effectively retain the knowledge obtained from the pretrained model. Inspired by this observation, we enforces a constraint for model update in CIL, which bounds the magnitude of weight updates from the base model at every $e_b$ epochs as

$$\Delta\Theta \leftarrow \begin{cases} B \cdot \frac{\Delta\Theta}{\|\Delta\Theta\|}, & \text{if } \|\Delta\Theta\| > B \\ \Delta\Theta, & \text{otherwise} \end{cases}, \tag{4}$$

where $\Delta\Theta$ denotes the displacement of the current model from the base model, $M_k^{\text{base}}(\cdot)$, and $B$ serves as the threshold for bounding the gradient magnitude. Note that the proposed bounded model update is performed within an incremental stage jointly with intra-task weight averaging. Given that the base model is presumed to hold the knowledge of previous tasks, this strategy aims to prevent the model from straying too far from the base model acquired in prior stages.

## 4 EXPERIMENTS

This section describes the datasets, evaluation protocol, and the implementation details of our algorithm. We then report the results of our algorithm on the standard benchmarks for CIL and conduct ablation studies to demonstrate the effectiveness of our method.

### 4.1 DATASETS AND EVALUATION PROTOCOL

We conduct CIL experiments on two datasets, CIFAR-100 (Krizhevsky et al., 2009) and ImageNet-100/1000 (Russakovsky et al., 2015). CIFAR-100 contains 50,000 training and 10,000 validation images in 100 classes. ImageNet-1000 contains 1,000 classes for 1.2M training and 50,000 validation images and ImageNet-100 is constructed with the first 100 classes of ImageNet-1000. For

Table 1: CIL performance (%) on CIFAR-100. The proposed training technique (BaA) consistently improves performance when plugged into the existing methods. Note that we run 3 experiments with 3 different orders and report the average result. Models with ∗ denotes the report of reproduced results. The bold-faced numbers indicate the best performance.

| | CIFAR-100 | | | |
|---|---|---|---|---|
| Number of tasks | 5 | 10 | 25 | 50 |
| iCaRL (Rebuffi et al., 2017) | 58.08 | 53.78 | 50.60 | 44.20 |
| BiC (Wu et al., 2019) | 56.86 | 53.21 | 48.96 | 47.09 |
| Mnemonics (Liu et al., 2020b) | 63.34 | 62.28 | 60.96 | - |
| GeoDL* (Simon et al., 2021) | 65.34 | 63.61 | 60.21 | 52.28 |
| UCIR (Hou et al., 2019) | 64.01 | 61.22 | 57.57 | 49.30 |
| PODNet* (Douillard et al., 2020) | 64.83±0.62 | 62.75±0.74 | 60.73±0.62 | 58.37±0.83 |
| PODNet (Douillard et al., 2020) + BaA | **67.69**±0.50 | **66.49**±0.50 | **64.93**±0.42 | **63.29**±0.37 |
| AFC* (Kang et al., 2022) | 66.11±0.60 | 64.77±0.74 | 63.68±0.74 | 61.94±0.60 |
| AFC (Kang et al., 2022) + BaA | **67.96**±0.70 | **67.10**±0.68 | **66.12**±0.67 | **65.38**±0.41 |
| FOSTER* (Wang et al., 2022) | 72.23±0.51 | 69.12±0.67 | 65.45±0.90 | 59.60±0.85 |
| FOSTER (Wang et al., 2022) + BaA | **73.03**±0.61 | **70.58**±0.64 | **66.79**±0.84 | **62.47**±0.97 |

Table 2: CIL performance (%) on ImageNet-100/1000. BaA demonstrates significant performance gains when integrated into existing methods, even in the large-scale benchmarks for CIL.

| | ImageNet-100 | | | ImageNet-1000 | |
|---|---|---|---|---|---|
| Number of tasks | 5 | 10 | 25 | 5 | 10 |
| iCaRL (Rebuffi et al., 2017) | 65.56 | 60.90 | 54.56 | 51.36 | 46.72 |
| BiC (Wu et al., 2019) | 68.97 | 65.14 | 59.65 | 45.72 | 44.31 |
| Mnemonics (Liu et al., 2020b) | 72.58 | 71.37 | 69.74 | 64.54 | 63.01 |
| GeoDL* (Simon et al., 2021) | 73.87 | 73.55 | 71.72 | 65.23 | 64.46 |
| UCIR (Hou et al., 2019) | 71.04 | 70.71 | 62.94 | 64.34 | 61.18 |
| PODNet* (Douillard et al., 2020) | 74.06 | 71.51 | 67.31 | 68.18 | 65.58 |
| PODNet (Douillard et al., 2020) + BaA | **75.98** | **74.08** | **70.70** | **69.53** | **67.76** |
| AFC* (Kang et al., 2022) | 76.91 | 75.26 | 73.65 | 68.06 | 66.39 |
| AFC (Kang et al., 2022) + BaA | **77.05** | **76.35** | **74.35** | **70.28** | **69.51** |
| FOSTER* (Wang et al., 2022) | 80.22 | 78.15 | 71.74 | – | – |
| FOSTER (Wang et al., 2022) + BaA | **80.59** | **79.20** | **73.24** | – | – |

fair comparisons, we follow three different class orders for CIFAR-100 while a class order specified in (Douillard et al., 2020) is used for ImageNet-100 and ImageNet-1000.

We incorporate the proposed training technique (BaA) into various state-of-the-art CIL algorithms based on knowledge distillation (PODNet (Douillard et al., 2020), AFC (Kang et al., 2022)), architecture expansion (FOSTER (Wang et al., 2022)), and virtual class augmentation (IL2A (Zhu et al., 2021)). Note that IL2A is employed to compare with non-exemplar-based methods.

Following the previous works (Douillard et al., 2020; Hou et al., 2019; Liu et al., 2021), we first train the model using a half of the classes in the initial stage and split the remaining classes into 5/10/25/50 stages for CIFAR-100, 5/10/25 stages for ImageNet-100, and 5/10 stages for ImageNet-1000 to simulate CIL scenarios. We test models on all the seen classes at each incremental stage, and report the *average incremental accuracy* (Rebuffi et al., 2017; Douillard et al., 2020; Hou et al., 2019)—average accuracy over all incremental stages.

## 4.2 IMPLEMENTATION DETAILS

As our approach is a plug-in method that can be incorporated into existing baselines, we follow the implementation settings of the existing methods (Douillard et al., 2020; Kang et al., 2022; Wang et al., 2022) in principle. We adopt ResNet-32 for CIFAR-100 and ResNet-18 for ImageNet as the backbone networks. We employ SGD with a momentum of 0.9. The hyperparameter setting including learning rates, batch sizes, training epochs, distillation loss weight, and herding strategies

Table 3: Component analysis of our algorithm for inter-task weight averaging, intra-task weight averaging, and bounded model update. We evaluate the impact of these components using the metrics of forgetting and average new accuracy, which reflect the balance between stability and plasticity in CIL, in addition to overall accuracy.

| | Inter-task | Intra-task | Bounded | Forgetting ↓ | Avg. new acc. ↑ | Overall acc.↑ |
|---|---|---|---|---|---|---|
| (a) BaA | ✓ | ✓ | ✓ | 15.38 | 59.35 | **65.38** |
| (b) w/o inter-task | | ✓ | ✓ | 21.77 | 62.74 | 61.81 |
| (c) w/o intra-task | ✓ | | ✓ | **13.10** | 51.60 | 65.08 |
| (d) w/o bounded | ✓ | ✓ | | 18.72 | **64.14** | 64.21 |

Table 4: CIL performance (%) with a limited memory budget on CIFAR-100: 1 exemplar memory per class for PODNet (Douillard et al., 2020) and AFC (Kang et al., 2022), and no memory for IL2A (Zhu et al., 2021). BaA demonstrates significant enhancement under strict memory constraint.

| | CIFAR-100 | | | |
|---|---|---|---|---|
| Number of tasks | 5 | 10 | 25 | 50 |
| PODNet* (Douillard et al., 2020)   (1 exemplar) | 43.70 | 34.19 | 26.58 | 14.78 |
| PODNet (Douillard et al., 2020) + BaA   (1 exemplar) | **52.27** | **45.63** | **36.81** | **20.84** |
| AFC* (Kang et al., 2022)   (1 exemplar) | 49.82 | 42.78 | 35.51 | 23.59 |
| AFC (Kang et al., 2022) + BaA   (1 exemplar) | **53.41** | **48.92** | **46.08** | **35.25** |
| IL2A* (Zhu et al., 2021)   (no exemplar) | 65.70 | 58.14 | 54.40 | 20.42 |
| IL2A (Zhu et al., 2021) + BaA   (no exemplar) | **67.46** | **61.80** | **58.25** | **43.54** |

is identical to the baseline algorithms. The memory budget size is set to 20 per class unless specified otherwise. Detailed description about hyperparameters for our method is illustrated in Section A

## 4.3 RESULTS ON CIFAR-100 AND IMAGENET-100/1000

**CIFAR-100** We plug in our method to PODNet (Douillard et al., 2020), AFC (Kang et al., 2022) and FOSTER (Wang et al., 2022). Following previous works (Douillard et al., 2020; Kang et al., 2022), we conduct three experiments with different class orders and report the mean and standard deviation. Table 1 illustrates that the proposed algorithm, denoted by BaA, enhances the performance of the baseline models in all CIL scenarios, with notable margins for various algorithms and numbers of tasks.

**ImageNet-100/1000** Table 2 shows the results on large-scale benchmarks, ImageNet-100/1000. While the proposed method consistently boosts the baseline algorithms, Table 2 clearly illustrates that the performance gains are particularly remarkable for ImageNet-1000, which is the most challenging benchmark for CIL. Additionally, across all datasets, the proposed algorithm demonstrates enhanced performance gains with an increasing number of tasks. This is partly because BaA is more robust to catastrophic forgetting, which typically worsens as the number of tasks increases. This characteristic is highly desirable for CIL in practical scenarios, which have no restrictions or information on the number of incoming tasks.

## 4.4 ABLATION STUDIES

We perform various ablation studies to validate the effectiveness of the proposed training technique. All the experiments are performed on CIFAR-100 with 50 incremental stages unless specified otherwise. To measure stability and plasticity, we compute the average new accuracy, which is the average accuracy of new classes over incremental stages, and the forgetting metric (Lee et al., 2019), which is the average of the performance degradation for each class.

**Variations of the proposed method** Table 3 presents the results obtained from different combinations of components in BaA. Each component contributes to the overall improvements, albeit in different ways. Comparing BaA to w/o inter-task and w/o bounded, we observe a decrease in forgetting, indicating that inter-task weight averaging and bounded updates effectively help retain previous knowledge. However, the low average new accuracy in w/o intra-task suggests that in-

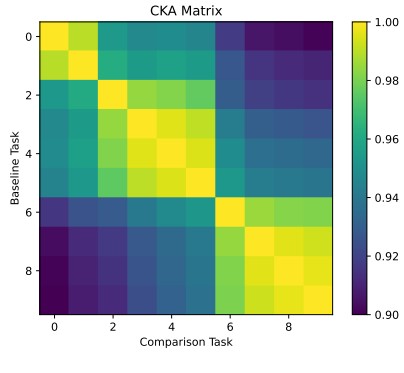
(a) CKA of AFC (Kang et al., 2022)

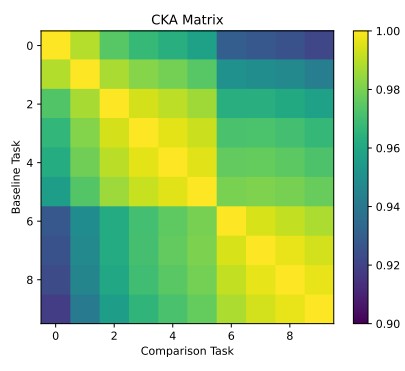
(b) CKA of AFC (Kang et al., 2022) + BaA

Figure 3: CKA between models after training individual incremental stages. We visualize the similarity between pairs of models obtained from two different tasks—baseline task and comparison task—by measuring CKA of the representations of test examples of all classes learned up to baseline task extracted from the two models.

tegrating bounded updates solely with inter-task weight averaging leads to significant degradation in adaptation. On the other hand, the increase in average new accuracy in ours compared to w/o intra-task indicates that the inclusion of intra-task weight averaging, in combination with bounded updates, helps mitigate this problem. Thus, the combination of these components enhances the performance of our method in retaining previous knowledge while facilitating adaptation to new tasks.

**Computational cost** To further substantiate the efficacy of our approach, we provide precise training time overhead induced by our method. Based on a single NVIDIA RTX-8000 GPU with a ResNet-32 backbone, we observe that the inter- and intra-task weight averaging only take 0.003 seconds each, whereas the bounded model update operation requires 0.011 seconds. Given that these operations only occur intermittently, *e.g.*, per task or every several epochs, the additional training time is marginal. Note that one epoch of additional forwarding is required for computing BN statistics per task. Because BaA is a training technique, it incurs no extra overhead for inference.

**Results on limited memory budgets** We evaluate the performance of BaA with exemplar-based methods such as PODNet (Douillard et al., 2020) and AFC (Kang et al., 2022) when only one exemplar is available. We also verify the effectiveness of BaA when it is combined with non-exemplar-based method, IL2A (Zhu et al., 2021). As shown in Table 4, our algorithm significantly outperforms existing methods when operating under limited memory budgets by exploiting model averaging techniques and constraining the amount of model updates. This property is desirable conceptually because CIL may have a large number of stages and even a small number of exemplars may be difficult to hold in practice.

**Similarity between representations** To comprehensively assess the impact of the proposed method on feature representations, we measure the similarity of representations. To this end, we extract the feature representations of the test examples from the final layer of the models trained on different tasks, *e.g.*, baseline task and comparison task using Centered Kernel Alignment (CKA) (Cortes et al., 2012; Kornblith et al., 2019) using test data of all classes learned up to the baseline task. Figure 3 clearly illustrates that BaA enhances feature similarities across the models trained in different incremental stages, which implies that BaA alleviates catastrophic forgetting.

**Variations in inter-task weight averaging** Table 5 shows the results from different strategies of inter-task weight averaging. Our moving average technique provides the same solution with the offline averaging, which outperforms Exponential Moving Averaging (EMA) with various smoothing factors. The EMA methods favor recent models and lead to forgetting the previous knowledge.

**Variations in intra-task weight averaging** We explore the characteristics of intra-task weight averaging by varying the weight averaging periods and the BN statistics computation strategies. According to Table 6, the proposed intra-task weight averaging technique is robust to the changes of averaging period but the results are affected by the methods to compute the BN statistics. Due to

Table 5: Analysis of inter-task weight averaging strategies. EMA denotes exponential moving average, where the numbers in parentheses indicate the smoothing factor; a higher smoothing factor assigns more weights to the most recent tasks.

|  | | Averaging factor | | |
| --- | --- | --- | --- | --- |
|  | Avg (Ours) | EMA (0.9) | EMA (0.5) | EMA (0.1) |
| Forgetting ↓ | **15.38** | 19.12 | 18.28 | 17.41 |
| Avg. new acc. ↑ | 59.35 | 61.34 | **61.74** | 59.10 |
| Overall acc. ↑ | **65.38** | 64.09 | 64.10 | 64.01 |

Table 6: Results by varying the weight averaging periods and the BN statistics computation methods in intra-class weight averaging. R means resetting the running statistics and NC indicates no change of the BN statistics after the weight averaging. † denotes our choice for reporting the main results.

|  | Weight averaging period | | | | BatchNorm | | |
| --- | --- | --- | --- | --- | --- | --- | --- |
|  | $1^\dagger$ | 5 | 10 | 15 | Ours | R | NC |
| Forgetting ↓ | **15.38** | 16.12 | 15.59 | 15.58 | **15.38** | 36.67 | 25.60 |
| Avg. new acc. ↑ | 59.35 | **59.41** | 58.50 | 58.95 | 59.35 | 39.75 | **60.47** |
| Overall acc. ↑ | 65.38 | 64.85 | 65.19 | **65.51** | **65.38** | 25.60 | 64.57 |

Table 7: Results by varying the bounding period and the size of the bound.

|  | Bounding period | | | | | Bounding threshold | | | |
| --- | --- | --- | --- | --- | --- | --- | --- | --- | --- |
|  | | | | | | 5 | $10^\dagger$ | 15 | 10 |
|  | 1 | 5 | 10 | $15^\dagger$ | w/o bounded | (w/ decay) | (w/ decay) | (w/ decay) | (w/o decay) |
| Forgetting ↓ | 15.73 | 15.89 | 16.45 | **15.38** | 18.72 | 16.53 | **15.38** | 16.33 | 18.09 |
| Avg. new acc. ↑ | 57.25 | 57.92 | 58.12 | 59.35 | **64.14** | 58.66 | 59.35 | 59.66 | **62.40** |
| Overall acc. ↑ | 64.76 | 64.81 | 64.73 | **65.38** | 64.21 | 64.50 | **65.38** | 64.99 | 64.44 |

the distinct characteristics of CIL that it should perform well on the previous classes, resetting the running statistics (R) gives severe performance degradation since the computed running statistics will be highly biased towards current tasks in computation procedure after the reset. Also, not forwarding the additional data path (NC) shows the degraded performance since there is discrepancy between the running statistics and intra-task averaged model since the statistics for the averaged model are not computed.

**Variations in bounded model updates** We conducted experiments with varying the frequency and the allowed size of the bounded model updates. As illustrated in Table 7, the overall accuracy is robust to the changes in the bounding frequency while applying the bounded model update results in clear advantage. For the bounding threshold, we observe that decaying the bounding threshold as the training proceeds in each stage gives improved results by reducing the forgetting. We argue that this phenomenon is attributed to the progressive convergence towards an optimal solution for the new task. As the learning process progresses, the updates gradually become more specialized and overfitted to the new task. This promotes the natural forgetting of previous information, making the decaying bounding threshold strategy valuable.

## 5 CONCLUSION

This paper introduces an innovative CIL method that utilizes weight ensemble techniques to handle the catastrophic forgetting. The method redefines CIL as a sequential transfer learning process and introduces inter-task and intra-task weight averaging, and bounded update techniques for enhanced model stability and adaptability.

The proposed strategy is seamlessly integrated into existing methods without modifications to the network architecture or loss functions. By leveraging the weights of previous models, our approach mitigates data dependency on preceding tasks, addressing concerns related to data privacy and memory budget.

**Reproducibility** We clarify all the hyperparameters needed for reproducing the proposed algorithm in Section 4.2 of main paper and A of appendix. We use the official code of each baseline for conducting experiments. We will release the code for reproducing the proposed method soon.

**Ethics statement** The proposed algorithm enhances the robustness and reliability of deep neural networks by resolving the catastrophic forgetting which can yield harmful impact under continuously shifting data distribution.

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

## A METHOD SPECIFIC HYPERPARAMETERS

Detailed description about hyperparameters deployed for our method is illustrated in Section A For CIFAR-100 and ImageNet-100 with PODNet (Douillard et al., 2020) and AFC (Kang et al., 2022), we perform intra-task weight averaging for every epoch, bounded updates for every 15 epochs and we decay the bounding threshold by $10 * (1 - \frac{\text{current epoch}}{\text{total epoch}})$. For ImageNet-1000 with PODNet (Douillard et al., 2020) and AFC (Kang et al., 2022), we perform intra-task weight averaging for every epoch, bounded updates for every epoch and we decay the bounding threshold by $30 * (1 - \frac{\text{current epoch}}{\text{total epoch}})$. In case of FOSTER (Wang et al., 2022), we perform intra-task weight averaging, bounded updates for every 10 epochs and we set the bounding threshold by 10. For IL2A (Zhu et al., 2021), we apply bounded updates for every 50 epochs and we decay the bounding threshold by $10 * (1 - \frac{\text{current epoch}}{\text{total epoch}})$. Since intra-task averaging enhances the adaptivity and degrades stability, we do not use intra-task averaging for IL2A (Zhu et al., 2021), which operates in non-exemplar scenario which suffers from extremely low stability.

## B ANALYSIS ON HYPERPARAMETERS

Table 8: Results by varying the bounding threshold ($B$) for CIFAR-100 with 50 incremental stages for AFC (Kang et al., 2022).

| Bounding Threshold | 5 (w/ decay) | 10 (w/ decay) | 20 (w/ decay) | 30 (w/ decay) | No bounding |
|---|---|---|---|---|---|
| Forgetting ↓ | 16.53 | **15.38** | 16.73 | 16.54 | 18.72 |
| Average New Accuracy ↑ | 58.66 | 59.35 | 59.71 | 59.82 | **64.14** |
| Overall Accuracy ↑ | 64.50 | **65.38** | 64.78 | 64.75 | 64.21 |

Table 9: Results by varying the bounding threshold ($B$) for ImageNet-1000 with 5 incremental stages for AFC (Kang et al., 2022).

| Bounding Threshold ($B$) | 5 (w/ decay) | 10 (w/ decay) | 20 (w/ decay) | 30 (w/ decay) |
|---|---|---|---|---|
| Forgetting ↓ | 4.11 | 3.75 | **3.34** | 3.54 |
| Average New Accuracy ↑ | 68.58 | 68.42 | **69.62** | 69.36 |
| Overall Accuracy ↑ | 69.47 | 69.53 | **70.55** | 70.28 |

Table 10: Comparison of performance w/ and w/o decay of bounding threshold for CIFAR-100 with AFC (Kang et al., 2022) after 10 and 50 incremental stages.

| | 10 steps | | 50 steps | |
|---|---|---|---|---|
| | w/ decay | w/o decay | w/ decay | w/o decay |
| Forgetting ↓ | **13.05** | 14.67 | **15.38** | 18.09 |
| Avg. New Accuracy ↑ | 66.09 | **68.82** | 59.35 | **62.40** |
| Overall Accuracy ↑ | **67.10** | 67.08 | **65.38** | 64.44 |

The method proves robust against the variations of bounded updates, always outperforming no bounding method as depicted in Table 8, 9 and 10. If needed, the selection of threshold and decay can be guided by the desired representation change. Table 8, 9 and 10 illustrate that larger thresholds and discarding decaying lead to higher average new accuracy. This is because a larger threshold and absence of decaying allows more change on representations. For complex tasks, e.g., ImageNet-1k, which demand larger changes at each stage, a model benefits from a large bounding threshold. Likewise, Table 10 demonstrates that the benefits of decaying are diminished as the incoming task grows in complexity, due to the increase in the number of classes per task. This situation arises because the

challenge of adapting to the new task becomes significant, in addition to retaining previous knowledge. In summary, one can select the bounding hyper-parameters by considering the complexity of the incoming task and the need for representation change, given the method's robustness.

