# OpenReview forum: "Bound and Average: Leveraging Weights as Knowledge for Class Incremental Learning"
_ICLR.cc/2024/Conference — ICLR 2024 Conference Withdrawn Submission_

### Official Review · Reviewer_m7u4 · 2023-10-22

**Soundness:** 2 fair
**Presentation:** 3 good
**Contribution:** 2 fair
**Rating:** 3
**Confidence:** 5

**Summary:**

This paper Bound-and-Average (BaA) for Class Incremental Learning which consists of weight ensemble and constrained optimization. The weighted ensemble contains the inter-task weight average and intra-task weight average. The inter-task weight average is proposed to maintain knowledge of previous stages and the intra-task weight average is proposed to enhance the learning of the current stage. Moreover, a bounded update technique is proposed to preserve knowledge. Experiments show that the proposed BaA can improve existing CIL methods.

**Strengths:**

The proposed method BaA is simple but effective, and the method can be integrated into existing methods to improve the performance without modifying their algorithm or architecture.

**Weaknesses:**

(1) The contribution of this paper appears to be insufficient to fully support the article. Similar EMA methods are widely used in self-supervised learning and some incremental learning methods [1,2].

[1]Yu C, Shi Y, Liu Z, et al. Lifelong person re-identification via knowledge refreshing and consolidation[C]//Proceedings of the AAAI Conference on Artificial Intelligence. 2023, 37(3): 3295-3303.

[2]Liang M, Zhou J, Wei W, et al. Balancing between forgetting and acquisition in incremental subpopulation learning[C]//European Conference on Computer Vision. Cham: Springer Nature Switzerland, 2022: 364-380.

(2) Lack of analysis experiments or visualization about how the proposed weight average mitigates the forgetting issue.

(3) Tables 1 and 2 show that BaA can improve the performance of certain CIL methods. Experiments on the latest CIL methods in 2023 should be done to show that the BaA can also be integrated into the SOTA methods.

**Questions:**

In Table 3, the results of the experiment are somewhat perplexing as the average new accuracy appears to be lower than the overall accuracy. In my assessment, the new accuracy represents the precision of classes of the current task, and the overall accuracy is expected to be lower than the average new accuracy due to knowledge forgetting.

---

### Official Review · Reviewer_eNr4 · 2023-10-26

**Soundness:** 2 fair
**Presentation:** 3 good
**Contribution:** 2 fair
**Rating:** 5
**Confidence:** 5

**Summary:**

The paper tackles the problem of class incremental learning by proposing weight-average techniques and bounded optimization. For weight averaging, both inter-task and intra-task weight ensembles are used. It was evaluated in conventional benchmarks with some improvements.

**Strengths:**

(1) The paper is clearly written and easy to follow.
(2) The online ensemble of old models is efficient.
(3) The method is integrated into three methods and the performance gain is clear.
(4) The ablation study of different components is thorough.

**Weaknesses:**

(1) The weight-average strategies are already used in some CL methods [A, B, C], it would be good to discuss the differences and the novelty compared to these existing methods.
(2) The section of Bounded model update seems just a direct application of the existing method, the novelty is limited. The performance improvement of this section shown in the experiments is only marginal (see Table 3 and Table 7).
(3) Some sentences are unclear. for instance, the Select() operation in Eq.2 is confusing; The threshold B in Eq. 4 is not dynamically determined but fixed, which sounds weird to me.
(4) More recent related work can be considered, especially papers published in 2023.
(5) Figure 3 shows the similarity between different models, and the lowest value is 0.9, It is unclear what it means for the two figures.

[A] Sarfraz, Fahad, Elahe Arani, and Bahram Zonooz. "Error Sensitivity Modulation based Experience Replay: Mitigating Abrupt Representation Drift in Continual Learning." ICLR 2023.
[B] Xiao, Jia-Wen, et al. "Endpoints Weight Fusion for Class Incremental Semantic Segmentation." CVPR 2023.
[C] Lin, Guoliang, Hanlu Chu, and Hanjiang Lai. "Towards better plasticity-stability trade-off in incremental learning: A simple linear connector." CVPR 2022.

**Questions:**

It would be good to discuss the questions in the Weaknesses Section.

---

### Official Review · Reviewer_yYZY · 2023-10-31

**Soundness:** 2 fair
**Presentation:** 3 good
**Contribution:** 2 fair
**Rating:** 3
**Confidence:** 5

**Summary:**

This paper proposes a new method called "Bound-and-Average (BaA)" for Class Incremental Learning (CIL). BaA integrates weight ensemble and constrained optimization, reducing the forgetting of knowledge from previous tasks when learning new ones. It contains two types of weight averaging: inter-task averaging and intra-task averaging. This approach requires no modifications to existing CIL architectures or learning objectives and demonstrates performance improvements on standard CIL benchmarks.

**Strengths:**

1) The approach is straightforward to understand. The introduction section clearly explains the motivation of the proposed method. The illustrations and discussions are clear.

2) Ablation studies effectively showcase how each part of the proposed method contributes to its overall performance.

**Weaknesses:**

1) The novelty of the method is limited. The idea of model averaging is simple, following many previous ensemble methods. Furthermore, the authors claim "which can be conveniently integrated into existing CIL methods", but there are already many methods that can similarly be incorporated into past models and have achieved good results [Xu H et al., 2020, Simon C et al., 2021, Hu X et al., 2021]. The proposed method is not compared with these methods.

2) There is a lack of comparison with state-of-the-art (SOTA) methods. From the experimental results in Table 1, when compared with newer saved sample methods, the performance improvement of this method has reduced significantly, dropping from 2.86% to 0.8%. However, Foster is not the current SOTA method. For example, [Yan S et al., 2021] and [Chen X et al., 2023] both perform much better than Foster. There is a lack of experimental results with BaA and these methods to prove its effectiveness.

3) The proposed method requires calculating the average value of all previous task models. This means it needs to store all the models from previous tasks, incurring significant amount of memory. Especially when compared with methods that do not save samples, this results in an unfair comparison.

[Xu H et al., 2020] Aanet: Adaptive aggregation network for efficient stereo matching, CVPR2020
[Simon C et al., 2021] On learning the geodesic path for incremental learning, CVPR2021.
[Hu X et al., 2021] Distilling causal effect of data in class-incremental learning, CVPR2021.
[Yan S et al., 2021] Dynamically expandable representation for class incremental learning, CVPR2021.
[Chen X et al., 2023] Dynamic Residual Classifier for Class Incremental Learning, CVPR2023.

**Questions:**

See Weaknesses.

---

### Official Review · Reviewer_vNSZ · 2023-11-01

**Soundness:** 2 fair
**Presentation:** 2 fair
**Contribution:** 1 poor
**Rating:** 3
**Confidence:** 3

**Summary:**

The submission proposes to tackle the class-incremental learning problem with inter-task and intra-task weight averaging. The former facilitates the incorporation of previous knowledge so as to counter catastrophic forgetting, while the latter enhances the performance of the current task model. They further include the bounded parameter update to regularize the parameter changes, which strengthens the effect of avoiding forgetting. They validate their method on multiple class-incremental datasets and achieve competitive results.

**Strengths:**

- The authors propose to utilize current weight averaging and bounded model update techniques to achieve a better balance between stability and plasticity.

**Weaknesses:**

- Lack of comparison with a highly relevant method. [1] also proposes to utilize the previous knowledge with ‘inter-task ensemble’, while enhancing the current task’s performance with ‘intra-task’ ensemble. Yet, the authors didn’t include the method comparison or performance comparison.

- Novelty is limited. From my perspective, the submission simply applies the existing weight averaging and the bounded update to the class incremental learning problems.

- No theoretical justification or interpretation. Is there any theoretical guarantee that to what extent inter-task weight averaging or bounded update counter catastrophic forgetting?

- Though it shows in Table 3 that bounded model update mitigates the forgetting, the incorporation of bounded model update doesn’t seem to have enough motivation from the methodological perspective. The inter-task weight average is designed to incorporate both old and new knowledge, which is by design enough to tackle forgetting.


##### References:

1.Miao, Z., Wang, Z., Chen, W., & Qiu, Q. (2021, October). Continual learning with filter atom swapping. In International Conference on Learning Representations.

**Questions:**

As shown in the weakness section.